# Calpain-1 and Calpain-2 in the Brain: What Have We Learned from 45 Years of Research?

**DOI:** 10.3390/cells14171301

**Published:** 2025-08-22

**Authors:** Michel Baudry, Xiaoning Bi

**Affiliations:** 1Department of Basic Medical Sciences, College of Dental Medicine, Western University of Health Sciences, Pomona, CA 91766, USA; 2Department of Biomedical Sciences, College of Osteopathic Medicine of the Pacific, Western University of Health Sciences, Pomona, CA 91766, USA; xbi@westernu.edu

**Keywords:** signaling pathways, phosphorylation, synaptic plasticity, neurodegeneration, neurogenesis, inhibitors, cytoskeleton

## Abstract

Although the calcium-dependent proteases, calpains, were discovered more than 60 years ago, we still know very little regarding their functions, mostly because very few studies are addressing questions related to specific members of this relatively large family of cysteine proteases. The “classical calpains”, calpain-1 and calpain-2, are ubiquitous and have received more attention because of the special roles they play in the brain. The authors have been studying the properties and functions of these two calpain isoforms in the brain for over 45 years, and this review will focus on what has been learned over this period of time. In particular, we will discuss the numerous studies that have led to the notion that calpain-1 and calpain-2 play opposite functions in the brain on processes ranging from neuronal survival or death, synaptic plasticity, and learning and memory to neurogenesis. Mechanisms underlying these opposite functions are starting to be understood and the findings support the notion that such opposite functions might be a general feature of these two isoforms in any type of cell. This review concludes with a discussion of the potential benefits of selective calpain-2 inhibitors for the treatment of a variety of neurological disorders.

## 1. Introduction

Much has been learned over the last 45 years regarding the properties and functions of various members of the family of the calcium-dependent proteases, calpains. These enzymes are found in all living organisms, reflecting their ancient evolutionary origins and essential roles in nearly all aspects of cellular biology. Since the discovery of calpains in 1964, these calcium-dependent, soluble cysteine proteases have been recognized as critical regulators of many cellular functions. Among them, calpain-1 and calpain-2—which are present in the brain—have emerged as key regulators of synaptic plasticity, neuronal survival and neurodegeneration, and neurogenesis. Initial biochemical characterizations in the late 1970s and early 1980s established that calpain-1 (also known as μ-calpain) and calpain-2 (also known as m-calpain) differ in their calcium requirements for activation, with calpain-1 responding to micromolar and calpain-2 to millimolar calcium concentrations. By the late 1980s, several studies demonstrated that calpains play a crucial role in cytoskeletal remodeling, particularly through the proteolysis of spectrin and other structural proteins, implicating them in activity-dependent synaptic reorganization.

The 1990s marked a turning point in our understanding of the functions of calpains in neurodegeneration as it was shown that prolonged calpain activation followed either seizure activity or traumatic brain injury (TBI), resulting in axonal and neuronal damage. The early 2000s brought significant advances in identifying specific substrates of calpain-1 and calpain-2, including two major types of ionotropic glutamate receptors—AMPA (α-amino-3-hydroxy-5-methyl-4-isoxazolepropionic acid) and NMDA (N-methyl-D-aspartate)—as well as several associated proteins that regulate their trafficking and insertion into postsynaptic membranes, directly linking calpain activity to learning and memory processes. Over the last 15 years, a number of studies have shown that calpain-1 activation is essential for long-term potentiation (LTP) of synaptic transmission by facilitating synaptic strengthening through proteolytic modification of key signaling proteins, while calpain-2 activation has an opposite function, reducing LTP magnitude and contributing to long-term depression (LTD) and synaptic weakening. Moreover, calpain-1 activation has been shown to be neuroprotective, while calpain-2 activation is directly associated with axonal degeneration and neuronal death. More recent studies have revealed that dysregulation of these proteases is a major contributor to neurodegenerative diseases, including cerebellar ataxia, Alzheimer’s, Huntington’s, and Parkinson’s diseases, as well as TBIs.

This review will explore the biochemical properties, activation mechanisms, and physiological roles of calpain-1 and calpain-2, focusing on key discoveries spanning five decades of research in the field. By examining the contributions of these two calpain isoforms to synaptic plasticity, learning and memory, and neurodegeneration, we aim to provide a comprehensive understanding of their distinct and opposite functions as well as their potential as therapeutic targets for neurological disorders.

## 2. The Early Period, 1978–1985: Focus on Synaptic Plasticity

The first report of the existence of a neutral, calcium-activated protease in rat brain was published in 1964 [1]. Surprisingly, it took another 16 years before two studies were published demonstrating the presence of calpain in muscles and identifying its endogenous inhibitor, calpastatin [2,3]. The following years saw an explosion of reports on the purification and properties of two forms of calpain with different requirements for activation by calcium. These were found in different organs, such as brain, muscle, liver, lens, and retina, as well as in different species, including rat, pig, and rabbit. Based on their calcium requirements, the two isoforms were originally named μ-calpain (now referred to as calpain-1), which is activated by micromolar concentrations of calcium, and m-calpain (now calpain-2), which requires millimolar calcium concentrations for activation. These two isoforms are heteromeric proteins composed of large homologous catalytic subunits of about 80 kDa and a shared small regulatory subunit (now known as calpain-4) of about 30 kDa. A high-molecular-weight endogenous inhibitor was also identified and named calpastatin—from calpain and statin, a suffix commonly used to indicate an inhibitor—with a molecular weight of 120 kDa. It was also proposed that during evolution, calpain originated from the fusion of genes coding for a thiol protease and a calcium-binding protein [4].

A number of substrates were also identified, including glial fibrillary acidic protein (GFAP), myelin, neurofilament protein, and, in the brain, a cytoskeletal protein named fodrin [5]. It was then realized that brain fodrin was identical to erythrocyte α-spectrin [6]. Interestingly, calpain activation results in partial truncation rather than complete degradation of all the substrate proteins, which strongly suggests that the function of this type of proteases is not simply degradative but is in fact a post-translational regulatory mechanism [7,8,9].

Our laboratory started to work on calpain in the late 1970s following the finding that glutamate receptors could be regulated by calpain. These studies showed that incubating synaptic membranes in the presence of calcium resulted in an increase in ^3^H-glutamate binding to its receptors [10]. Interestingly, the effect of calcium could be observed when membranes were first incubated with calcium, followed by calcium wash-out prior to the binding assay, indicating that the calcium treatment had resulted in an irreversible change in membrane properties. While several calcium-dependent processes could have been involved, the report by the Murachi laboratory of the presence of calpain in the brain led to the investigation of the possible role of calpain in this process. In a series of studies, the effect of calcium on ^3^H-glutamate binding in rat hippocampal membranes was found to be the result of calpain activation [11,12,13]. It was also determined that calpain-1 was present in hippocampal membranes [14] and brain fodrin (spectrin) was identified as a calpain substrate [5], confirming the results previously found in erythrocyte membranes [15].

Parallel studies in the laboratory of Gary Lynch indicated that the induction of LTP of synaptic transmission elicited by brief trains of high-frequency stimulation at hippocampal synapses was critically dependent on an influx of calcium [16]. All these results led to the hypothesis that calpain activation plays a critical role in synaptic plasticity and in learning and memory [17]. This hypothesis was further supported by the finding that a thiol protease inhibitor, leupeptin, could prevent olfactory discrimination learning in rats [18].

Interestingly, during this period, a report indicated that calpain-2 could be co-purified with the cAMP-dependent protein kinase (PKA), which can phosphorylate calpain-2 as well as other substrates [19]. The significance of this finding would not be realized until almost 30 years later (see below). Similarly, the calcium dependency for calpain-2 activation was found to be dramatically reduced in the presence of certain types of phospholipids, in particular phosphatidylinositol [20,21]. Furthermore, this effect was specific for the autolysis of calpain-2, resulting in a prolonged calpain-2 activation. The findings from this period were extensively reviewed in Zimmerman and Schlaepfer [22], which stressed the signature of this class of proteases, that is, the partial proteolytic processing of their substrates, thus establishing calpains as a unique post-translational regulatory mechanism.

## 3. Calpain and Neurodegeneration: 1985–1994

In the mid-1980s, the focus of calpain research in the brain turned to neurodegeneration. This was the result of a number of key findings, which raised the possibility that calpain activation was not limited to the regulation of physiological processes but could play a critical role in neurodegeneration. First, calpain-1 expression was found to be prominent in several structures in which degeneration occurs spontaneously, e.g., in spinal motoneurons and olfactory neurons. In addition, calpain was found in regions susceptible to age-related pathologies, including cerebellar Purkinje cells, substantia nigra, and subiculum [23]. This distribution suggested that calpain-1 could be involved in both normal and pathological neuronal death. Second, calpain activation was found to be related to excitotoxicity-induced neuronal damage [24,25], or following lesion-induced neurodegeneration in the dentate gyrus [26,27]. Third, calpain inhibitors were found to be neuroprotective following ischemic episodes in both in vitro and in vivo experiments [28,29,30,31,32]. Fourth, several evidence linked calpain and Alzheimer’s disease (AD). Calpain was found to be localized in senile plaques and in neurons exhibiting neurofibrillary degeneration in AD [33,34]. Calpain was also found to be involved in the proteolysis of the ß-amyloid precursor protein (APP) [35]. The widespread activation of calpain in the brain of patients with AD led to the proposal that calpain represented a molecular mechanism for neurodegeneration in the disease [36]. All these findings led Gary Lynch, Carol Cotman, and Ralph Bradshaw from UC Irvine to create a biopharmaceutical company, Cortex Pharmaceuticals (Cortex), to develop calpain inhibitors for the treatment of neurodegeneration. Cortex did develop several novel calpain inhibitors [37], which were found to protect neurons following focal ischemia [38] and attenuate motor and cognitive deficits in a rat model of TBI [39]. These findings supported the notion that calpain was a novel target to treat acute neurodegenerative disorders [40]. However, Cortex did not further pursue the clinical development of calpain inhibitors for the treatment of neurological disorders for reasons that only became clear about 15 years later (see below).

Nevertheless, parallel findings continued to support a critical role for calpain activation in synaptic plasticity and learning and memory. Thus, hippocampal LTP was reduced by calpain inhibitors both in vitro and in vivo [41,42,43,44]. Olfactory discrimination learning was blocked by a calpain inhibitor [18].

These multiple functions of calpain provided a confusing picture regarding the underlying mechanisms and, in particular, the processes involved in calpain activation under physiological or pathological conditions. While it was tempting to use the different calcium sensitivities of calpain-1 and calpain-2 to account for the selective activation of calpain-1 and calpain-2 in these conditions [17], there was no convincing evidence to support this idea. Increasing the confusion regarding these different functions were the findings related to changes in calpain activity with aging. Thus, calpain activity was found to increase with age [45], and a comparative study of brain calpain activity suggested that it could be an important element regulating longevity in mammalian species, when it was observed that brain calpain activity was highly negatively correlated with maximal lifespan across various species [46,47].

## 4. Calpain and Glutamate Receptors: 1995–2012

The initial identification of calpain substrates focused on cytoskeletal proteins, including spectrin [5], neurofilament proteins [15], tubulin and microtubule-associated proteins (MAPs) [48], as well as desmin and vimentin [49]. Additional substrates include various protein kinases, such as calcium/calmodulin-dependent protein kinase IV (CaM KIV) [50] and CaMKII [51], protein kinase C [52], glycogen synthase kinase 3 (GSK-3) [53], as well as various phosphatases [54,55]. AMPA receptors were then found to be cleaved by calpain in adult rat brain in their C-terminal domains [56,57] and, in particular, the GluA1 subunits of AMPA receptors [58]. Furthermore, NMDA receptors were regulated by calpain-mediated proteolysis, a mechanism that was extensively studied by the group of David Lynch [59,60,61,62,63,64]. In addition to partially proteolyzing glutamate receptors, calpain was found to degrade several postsynaptic proteins involved in anchoring and stabilizing glutamate receptors in postsynaptic densities. These included PSD-95 (postsynaptic density protein 95) [65], glutamate-receptor-interacting protein (GRIP) [66], and stargazin [67]. Interestingly, the state of phosphorylation of glutamate receptors was found to be an important regulator of calpain-mediated truncation of the receptors. Thus, phosphorylation of AMPA and NMDA receptors either reduced [60] or enhanced their susceptibility to calpain cleavage [68], while fyn-mediated phosphorylation protected AMPA receptors from calpain cleavage but enhanced calpain-mediated cleavage of NMDA receptors [69]. In contrast, src-mediated phosphorylation reduced NMDA receptor cleavage and did not affect AMPA receptor cleavage [69]. Such complex interactions between the phosphorylation states of calpain targets have now been reported for tau [70], the activator of the cyclin-dependent kinase 5 (cdk5), p35 [71], spectrin [72], the transcription factor C/EBP [73], and GSK3ß [74]. This should not be too surprising since calpain interactions with its substrates rely on their 3D structures, which are clearly sensitive to their state of phosphorylation.

In addition to cleaving ionotropic glutamate receptors, calpain was found to cleave the metabotropic glutamate receptor, mGluR1α, in its C-terminal domain [75]. In this particular case, the resulting effect of the cleavage was quite dramatic as it switched the function of mGluR1α from being neuroprotective to neurodegenerative. This step was found to be a key step in excitotoxicity [76]. These findings led to the development of a small cell-permeable decoy peptide that could prevent calpain-mediated truncation of the receptors and be neuroprotective under ischemic or excitotoxic conditions [76].

These complex interactions between phosphorylation mechanisms and calpain-mediated proteolysis are proposed to play important roles in a number of physiological processes, such as synaptic plasticity [77], cell migration [78], and the remodeling of cytoskeletal anchorage complexes [79]. They are also proposed to participate in several pathological processes, including neurodegeneration [80]—especially AD [81]—cerebral vasospasm [82], and muscle dysfunction [83].

During this period, molecular cloning spearheaded by the group of Hiroyuki Sorumachi led to the identification of the calpain family, which consists of 16 members encoded by 16 different genes, *CAPN*1-16, although *CAPN4* is now designated as *CAPNS1* [84,85,86,87]. Based on the organization of different domains of the family members, calpains have been classified as classical and non-classical calpains. While some of the members are tissue-specific—such as calpain-3, which is only expressed in skeletal muscles—many are ubiquitously expressed [88]. Structural studies have helped in understanding the mechanism of activation by calcium, as well as the fact that these enzymes prefer to proteolyze unstructured regions of their target proteins due to the steric constraints of their catalytic sites. In particular, the group of Peter Davis reported the mechanism of calpain activation following the binding of 10 calcium ions to calpain, resulting in a conformational change and the opening of the catalytic site between the PC1 and PC2 domains of calpain (Figure 1) ([89], see also https://www.igakuken.or.jp/calpain/MainPages/ResearchTheme.html (accessed on 14 August 2025)). The catalytic site consists of a ubiquitous triad of amino acids (Cys105/His262/Asn286), responsible for the cysteine protease designation of calpains.

It is also interesting to note that other members of the calpain family were linked to specific diseases; thus, loss-of-function mutations in calpain-3 were shown to be responsible for limb-girdle muscular dystrophy 2A [90,91]. Down-regulation of calpain-9 has been linked to gastric cancer [92].

Increasing evidence has linked calpain-1 and calpain-2 to stroke, TBI, and AD [36,93,94,95,96,97], spinal cord injury [98], Parkinson’s disease (PD) [99], and global neurodegeneration [100].

## 5. Opposite Functions of Calpain-1 and Calpain-2 in the Brain: 2012–2025

The high calcium requirement for activation of both calpain-1 and calpain-2 has long been a puzzle for scientists working in this field since it exceeds by far the resting levels of intracellular calcium, and even those likely to be present following cellular activation. It has been widely accepted that this requirement ensures that calpain activation could take place only near sources of calcium influx from extracellular or intracellular stores, resulting in a temporally and spatially limited stimulation of this potentially damaging proteolytic system. Importantly, phosphorylation of various residues in calpain-1 and calpain-2 was found to regulate their activity. Mitogen-activated protein kinase (MAP Kinase)-mediated phosphorylation of serine 50 was found to activate both calpains [101,102], while PKA-mediated phosphorylation of serine 369—which, as mentioned earlier, had been reported in 1985—resulted in calpain-2 inhibition [103,104]. While the mechanisms underlying these effects are not clearly understood, it is assumed that these phosphorylation events modify the interactions of calpain-1 and calpain-2 with cell membranes or other proteins.

In cultured hippocampal neurons, the brain-derived nerve growth factor (BDNF) stimulated calpain-2 but not calpain-1, an effect that was blocked by MAPK inhibitors. BDNF-induced calpain-2 activation was preferentially localized in dendrites and dendritic spines of hippocampal neurons and was associated with actin polymerization, which was prevented by calpain inhibition [102]. BDNF has been shown to stimulate local dendritic protein synthesis [105], and calpain-2-mediated PTEN (phosphatase and tensin homolog) truncation was found to be involved in BDNF-mediated stimulation of this local protein synthesis [106]. Conditional deletion of CAPNS1 in the brain, which disrupts both calpain-1 and calpain-2 activity, resulted in altered dendritic morphology and LTP impairment in hippocampus, thus confirming a critical role for calpain in LTP [107]. Activation of synaptic NMDA receptors has been shown to be neuroprotective by stimulating the Akt survival pathway, while activation of extrasynaptic NMDA receptors is linked to a neurodegenerative pathway [108,109]. In cultured hippocampal neurons, activation of synaptic NMDA receptors resulted in the stimulation of calpain-1 and the cleavage of two splice variants of PH domain and Leucine-rich repeat Protein Phosphatase 1 (PHLPP1)—PHLPP1α and PHLPP1β—which inhibit the Akt pathway [110]. In contrast, extrasynaptic NMDAR activation had no effect on PHLPP1 and the Akt pathways but resulted in calpain-2-mediated degradation of striatal-enriched protein tyrosine phosphatase (STEP) and neuronal death [110]. This finding provided the first evidence that calpain-1 and calpain-2 play opposite functions in the brain. The suprachiasmatic nucleus circadian oscillatory protein (SCOP)—also known as PHLPP1β, a negative regulator of the extracellular signal-regulated kinase (ERK)—has been shown to play an important role in learning and memory as its overexpression blocks long-term memory encoding [111]. Calpain-1-mediated SCOP degradation was found to be involved in LTP induction, while calpain-2-mediated stimulation of mTOR (mechanistic target of rapamycin)-dependent SCOP synthesis restricted the magnitude of potentiation during early consolidation [112]. Thus, opposite effects of calpain-1 and calpain-2 control the formation and extent of LTP in hippocampus. These results were confirmed by finding that LTP induction was absent in calpain-1 knock-out (ko) mice [113], while LTP magnitude was enhanced by a selective calpain-2 inhibitor [112]. The neuroprotective properties of calpain-1 were also further supported by studies in mice, dogs, and humans with null mutations or deletions of calpain-1 [114,115], which result in cerebellar ataxia. Detailed studies in mice revealed that a lack of calpain-1 was associated with increased apoptosis throughout the brain during the early postnatal period [115]. As there are no selective calpain-1 inhibitors, our understanding of the functions of calpain-1 is limited to the findings obtained by studies with the calpain-1 ko mice, which could be biased due to the long-term changes in the expression of a number of genes affected by calpain-1 deletion [116]. Conversely, the neurodegenerative properties of calpain-2 were confirmed by a wide range of studies, demonstrating the critical role of calpain-2 in several animal models of neurodegeneration, including acute glaucoma [117], TBI [118], repeated mild concussion [119], and seizure-induced neuropathology [120].

All the data are, therefore, consistent with the notion that, in the brain at least, calpain-1 and calpain-2 play opposite functions in synaptic plasticity and learning and memory as well as in neuroprotection/neurodegeneration [121,122]. Obviously, a critical question concerns the potential mechanism(s) that could account for these opposing functions. One of the mechanisms we identified is related to the different targets for calpain-1 and calpain-2. Thus, calpain-1 activation results in the cleavage and inactivation of SCOP, a negative regulator of ERK, and PI3-Akt, a pro-survival kinase [112,123]. On the other hand, calpain-2 degrades and inactivates STEP, resulting in the activation of p38 and downstream cell death signaling pathways [110,124,125]. Another mechanism we identified was provided by the existence of different (PSD-95/Dlg/ZO-1) (PDZ) binding domains in the C-terminal domains of calpain-1 and calpain-2 [121]. We postulated that these domains result in the association of calpain-1 and calpain-2 with different signaling pathways—involved in neuroprotection and synaptic plasticity for calpain-1, and in opposing synaptic plasticity and stimulating neuronal damage for calpain-2 [88,122,126] (Figure 2). Calpain-1 is likely to be associated with PSD-95, which also binds the NMDA receptors, as well as neuronal nitric oxide synthase (nNOS) and PHLPP1 [127] (Figure 2A). Interestingly, all the members of the calpain family exhibit a PDZ binding domain in their C-terminal domains, which suggests that they are all associated with distinct signaling pathways [88]. In searching for a PDZ binding partner for calpain-2, we identified tyrosine phosphatase, PTPN13—also known as Fas-Associated Protein 1 (FAP1)—a negative regulator of apoptosis [128]. Importantly, PTPN13 is selectively cleaved and inactivated following calpain-2 activation, resulting in the activation of downstream signaling pathways, including apoptosis and various protein kinases, such as c-Abl (Figure 2B). This pathway could account for the abnormal tau phosphorylation observed following TBI [128]. Moreover, a fragment of PTPN13 with a molecular weight of about 130 kD leaks into the blood and its levels were found to be correlated with the degree of injury following TBIs in both mice and humans [129]. Interestingly, PTPN13 has five PDZ binding domains and also binds PIP2 and PTEN in addition to calpain-2 [130,131,132] (Figure 2B). This could account for the activation of calpain-2 following stimulation of extrasynaptic NMDA receptors—resulting in the truncation of PTEN and PTPN13—as well as in the autolysis of calpain-2—resulting in the prolonged activation of calpain-2—which has been shown following TBI or status epilepticus [118,120].

Many studies have shown that calpains participate in neurogenesis in adult mammalian brain [133,134,135,136,137]. As mentioned above, calpains play an important role in cell proliferation, survival, migration, and differentiation [135,137,138,139,140,141,142,143,144,145,146]. Participation of calpains in these processes is the result of their interactions with and cleavage of proteins involved in cell cycle regulation [142,144,147]—in particular cdk5 [148], which requires the binding of its regulatory subunit, p35. Calpain cleaves p35 into p25, which increases cdk5 activity and changes its substrate selectivity and subcellular localization. For a long time, the contributions of calpain-1 and calpain-2 in neurogenesis have been difficult to evaluate due to the lack of selective inhibitor or of mutant mice with selective down-regulation of calpain-1 or calpain-2. Calpain-1 is highly expressed in neural stem cells (NSCs) and immature neural progenitor cells (NPCs), while calpain-2 expression increases during neuronal differentiation [133,136,149], suggesting that calpain-1 and calpain-2 play different roles in neurogenesis. Using calpain-1 ko mice, we found that the proliferation of newborn cells was decreased in the dentate gyrus [150]. This decrease was comparable to that found in the dentate gyrus of the CAPNS1-ko mice [107,133]. These results suggest that calpain-1 plays a positive role in neurogenesis. In a very recent study, the transcription factor Myeloid Ecotropic Viral Integration Site 2 (MEIS2) was found to be a target of calpain-2 but not of calpain-1. Acute treatment of mice with a selective calpain-2 inhibitor, NA-184, resulted in increased MEIS2 levels in various brain regions, including the subventricular zone and the dentate gyrus, leading to an increase in neurogenesis [151]. Therefore, calpain-2 appears to be a negative regulator of neurogenesis. These findings further underscore the opposite functions of calpain-1 and calpain-2 in the brain.

## 6. Development of Selective Calpain-2 Inhibitors for the Treatment of Neurodegenerative Disorders

Calpains have been implicated in many disorders since their initial discovery. This topic has been covered by many interesting reviews throughout the years, and only a few are cited here due to space limitations [152,153,154,155,156,157]. In particular, the classical calpains have long been implicated in neurodegeneration broadly [158,159] and, more specifically, in stroke [160,161] and TBI [162,163]. Accordingly, numerous studies have tested calpain inhibitors to attenuate neurodegeneration in both stroke and TBI [163,164]. Some first-generation inhibitors showed promising effects in TBI models [165], though others failed to demonstrate efficacy [166,167]. As discussed previously, some of the inhibitors generated by Cortex Pharmaceuticals were also tested in animal models of neurodegeneration [38]. For example, administration of the keto-amide inhibitor, AK295, 15 min post-lateral fluid percussion injury reduced cognitive and motor impairments [39] but did not affect neuronal damage or calpain-mediated spectrin degradation [168]. Similarly, pre-injury administration of the non-selective calpain inhibitor, MDL-28710, mitigated short-term diffuse axonal injury in a rat impact acceleration model [169] and provided neuroprotection in the corpus callosum following fluid percussion injury. Notably, neuroprotection was limited to a 4-h window, though repeated dosing extended efficacy [170]. However, MDL-28710 did not prevent axonal transport deficits after stretch injury [171]. Even more advanced, BBB-permeable inhibitors, such as SNJ-1945 and MDL-28170, failed to provide sufficient efficacy or a clinically relevant therapeutic window in controlled cortical impact (CCI) models [165,166].

These outcomes may reflect limitations in specificity, potency, and isoform selectivity [172], as well as an incomplete understanding of the differential roles of calpain isoforms in the brain. As discussed above, calpain-1 and calpain-2 have now been found to exhibit opposing functions in synaptic plasticity and neurodegeneration [121]. Specifically, calpain-1 is required for theta-burst-induced LTP and is neuroprotective, while calpain-2 restricts LTP and promotes neurodegeneration [110,111,112]. The non-selectivity of earlier inhibitors likely masked these divergent effects and hindered their therapeutic development. Therefore, it is the balance between calpain-1 and calpain-2 activity that is responsible for many critical processes regulating cellular homeostasis in neurons and other cell types (Figure 3).

These findings imply that selective calpain-2 inhibitors could be extremely beneficial in a number of both acute and chronic neurological disorders, including TBI and ischemic damage, as well as AD and PD [173,174].

Although the first selective calpain-2 inhibitor we identified, NA-101 (Figure 4), was found to selectively bind to the catalytic triad [175] and exhibited favorable pharmacological characteristics, it was not suitable for clinical advancement. OSIRIS Data Warrior analysis yielded a drug-likeness score of -13.6, well below the threshold for typical drug candidates. Furthermore, NA-101 displayed a non-monotonic (inverted U-shaped) dose–response curve in CCI models, where low doses were neuroprotective and high doses worsened injury [118].

Consequently, we initiated a medicinal chemistry campaign to develop improved analogs. NA-101 shares a scaffold with other calpain inhibitors, including those in clinical development (e.g., ABT-957 by AbbVie; SNJ-1945 by Senju), as well as with clinically tested ketoamide serine protease inhibitors such as telaprevir and boceprevir (Figure 4). NA-101 is a peptidyl α-ketoamide that forms a reversible covalent bond with the calpain catalytic triad shown in Figure 1. Among reversible covalent calpain inhibitors, α-ketoamides offer particularly stable electrophilic warheads [177,178], making them promising leads.

Molecular dynamics simulations revealed that NA-101 adopts different binding poses in calpain-1 and calpain-2 due to isoform-specific differences in domain 3. Notably, calpain-1 lacks the hydrogen bond interaction between catalytic histidine and the P1′ methoxy group seen in calpain-2 [175]. This suggested that isoform selectivity could be improved through targeted substitutions at the P3 and P1′ positions. To guide structure-based optimization, we employed SILCS (Site Identification by Ligand Competitive Saturation) simulations [179], which generate fragment probability maps (FragMaps) of protein surfaces. At the P1′ site, calpain-1 favored substitutions at the para-position of the phenyl ring, while calpain-2 favored meta-position substitutions and accommodated additional H-bond donors/acceptors at the ortho-position.

Using this information, we synthesized ~130 analogs and screened them for selectivity toward human calpain-1 and calpain-2. Two compounds, NA-112 and NA-184, emerged as lead candidates based on their low Ki for calpain-2 and excellent isoform selectivity (Figure 4). Substitutions at the P3 site did not improve selectivity, likely due to its conformational flexibility. However, replacing the carbamate linker with a urea linker at this site enhanced both binding stability and metabolic resistance.

Importantly, both NA-112 and NA-184 cross the blood–brain barrier (BBB) and achieve >20-fold selectivity for calpain-2 over calpain-1 in vivo. In CCI models of TBI, both compounds provided robust neuroprotection and reduced circulating levels of the calpain-2 biomarker P13BP at 24 h post-injury [180].

NA-184 is a potent in vivo inhibitor of mouse calpain-2 with an IC50 of 130 nM and an ED50 of 0.13 mg/kg for neuroprotection in CCI models, with no calpain-1 inhibition up to 10 mg/kg. Against human calpain-2, NA-184 has an IC50 of 1.3 nM and does not inhibit calpain-1 at concentrations up to 10 µM.

As mentioned above, NA-184 has an additional property that makes it an interesting drug to address age-related neuronal disorders as it stimulates neurogenesis through increased levels of the transcription factor MEIS2 [151]. Stimulating neurogenesis—particularly in the adult hippocampus and subventricular zone—can offer several potential benefits for age-related neuronal disorders such as AD, PD, and age-associated cognitive decline. Hippocampal neurogenesis is closely linked to learning, memory, and pattern separation [181]. Aging and neurodegenerative diseases are associated with reduced neurogenesis, which correlates with cognitive deficits [182]. Stimulating neurogenesis could enhance neuronal survival, replenish neuronal populations, and improve synaptic plasticity, potentially restoring memory and learning abilities. New neurons can integrate into existing neural circuits and may help rebuild damaged networks [183]. This type of plasticity is particularly relevant for diseases like AD, where synaptic loss and network disruption are hallmarks. Moreover, neurogenesis is often accompanied by increased levels of neurotrophic factors such as BDNF and IGF-1 (insulin-like growth factor 1) [184]. These factors promote neuronal survival, reduce inflammation, and support the health of existing neurons—providing a neuroprotective environment. Finally, depression is commonly comorbid with neurodegenerative diseases, and hippocampal neurogenesis is implicated in the antidepressant response [185]. Enhancing neurogenesis might alleviate mood disturbances, improving quality of life and cognitive performance indirectly. Chronic inflammation impairs neurogenesis and contributes to cognitive decline. Stimulating neurogenesis may counteract neuroinflammation, either directly or indirectly by modulating microglial and astrocytic responses [186]. Interestingly, a recent study appears to confirm that neurogenesis is present in the dentate gyrus of adult humans [187], suggesting that stimulating neurogenesis in humans could have broad beneficial effects.

Altogether, these findings strongly support the development of specific calpain-2 inhibitors for the treatment of a variety of neurological disorders. The coming years should provide clinical validation to this hypothesis as clinical trials of NA-184 for TBI treatment are expected to be initiated in 2026.

## 7. Conclusions and Future Directions

Looking back at the state of knowledge on calpains in the early 1980s, it is clear that much has been learned regarding all the different features of calpain-1 and calpain-2. We know how these enzymes are activated by brief changes in calcium concentrations and/or phosphorylation status, a large number of substrates have been identified, and their participation in many important neuronal functions has been elucidated. A critical feature of these two isoforms is the existence of different PDZ binding domains, resulting in their association with different signaling pathways, which are mostly responsible for their opposite functions. While calpain-1 is required for hippocampal LTP and learning and memory, calpain-2 limits the magnitude of LTP and the extent of learning and memory. Similarly, calpain-1 is neuroprotective and stimulates neurogenesis, while calpain-2 is neurodegenerative and inhibits neurogenesis. These findings have led to the identification of selective calpain-2 inhibitors, which have been found to facilitate learning and memory and limit the extent of neuronal damage in several animal models of acute injury, including glaucoma and mild and severe concussions. They have also been found to stimulate neurogenesis in adult mouse brain. Several of these inhibitors are in the final stage of pre-IND studies and are scheduled to enter clinical trials for the treatment of various neurological disorders. The next few years should provide definitive answers regarding the successful translation of the pre-clinical studies to human therapeutic indications.

## 8. Limitations

We realize that we might have missed important studies in our literature search, and we apologize to the authors of these studies. It is also the case that we focused on the roles of calpain-1 and calpain-2 in the brain since there is still little information regarding the specific roles of these two calpain isoforms in other tissues and organs. It is our hope that this review will encourage scientists to expand on our studies in order to provide more information on this topic, as well as on the roles of the other members of the calpain family, which are still understudied.

## Figures and Tables

**Figure 1 cells-14-01301-f001:**
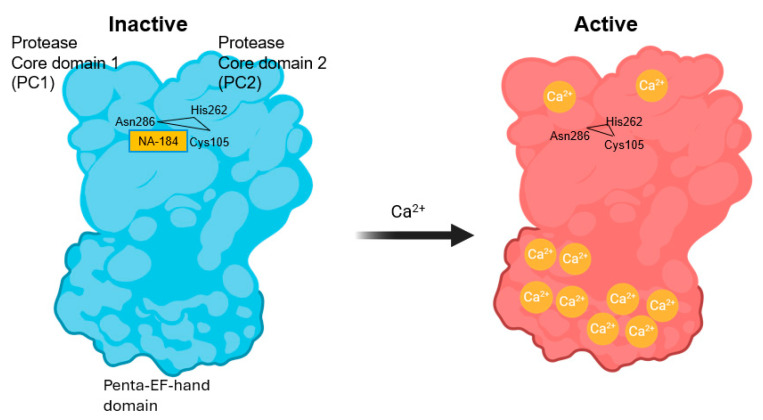
Schematic representation of the activation of calpain by calcium. In the absence of calcium, calpain-1/2 is in an inactive state and the catalytic triad located between Protease Core domain 1 (PC1) and Protease Core domain 2 (PC2), Cys105/His262/Asn286, is not functional (**left**); this panel also shows the binding sites of NA184. Upon binding of 10 calcium ions to various sites of the Penta–EF–hand domains and PC1 and PC2 domains, the catalytic triad becomes functional (**right**).

**Figure 2 cells-14-01301-f002:**
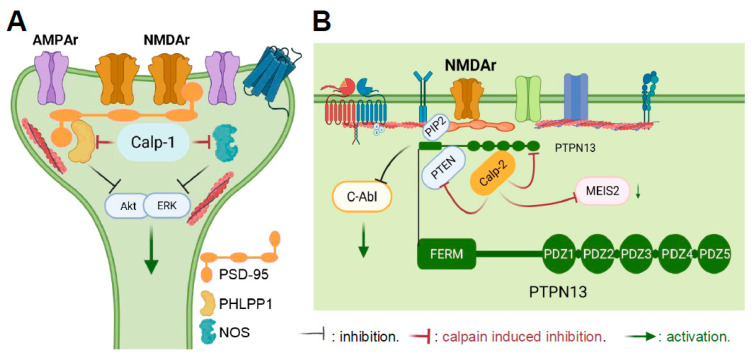
Different subcellular localizations and key actions of calpain-1 and calpain-2 in neurons. (**A**) Subcellular localization of calpain-1. Calpain-1 is downstream of synaptic NMDA receptors and is in a complex of proteins linked to PSD-95, which includes PHLPP1 and NOS. By cleaving PHLPP1, calpain-1 activation results in ERK and Akt activation, which are both involved in synaptic plasticity and in neuronal survival. (**B**) Subcellular localization and major actions of calpain-2. Calpain-2 is downstream of extrasynaptic NMDA receptors and is in a complex of proteins linked to PTPN13, which includes PTEN. Calpain-2 activation results in the cleavage of PTEN and of PTPN13, leading to stimulation of dendritic protein synthesis and c-Abl activation. This pathway is involved in both synaptic plasticity and in neurodegeneration. Calpain-2 also cleaves MEIS2, leading to reduced neurogenesis.

**Figure 3 cells-14-01301-f003:**
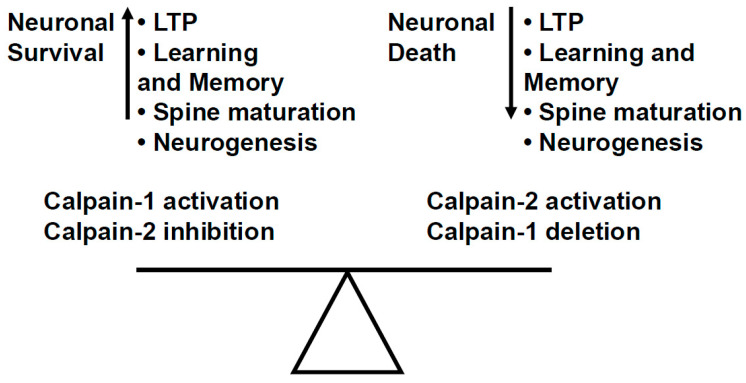
The balance between calpain-1 and calpain-2 activity plays a critical role in neuronal homeostasis. Calpain-1 activation and calpain-2 inhibition result in enhanced LTP and learning and memory, enhanced dendritic spine maturation, neuronal survival, and enhanced neurogenesis. On the other hand, calpain-2 activation and calpain-1 deletion/inhibition result in opposite functions, i.e., neuronal death, decreased LTP and learning and memory, inhibited spine maturation, and inhibited neurogenesis.

**Figure 4 cells-14-01301-f004:**
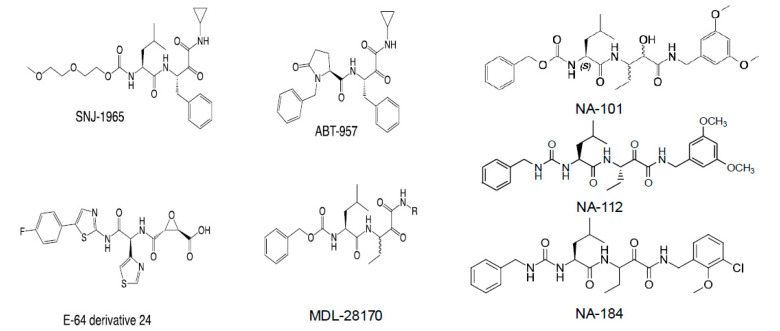
Structures of different calpain inhibitors. While SNJ-1965, ABT-957, E-64 derivative 24, and MDL-28170 are not selective for calpain-1 or calpain-2, NA-101, NA-112, and NA-184 are selective for calpain-2 over calpain-1. Note that ABT-957 was tested in a Phase I clinical trial for Alzheimer’s disease. While no significant side-effects were observed, AbbVie decided not to pursue the clinical development due to the lack of observable pharmacological effects at the doses tested [176].

## Data Availability

No new data were created or analyzed in this study.

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
