# Peer review of "Calpain-1 and Calpain-2 in the Brain: What Have We Learned from 45 Years of Research?"

_cells, 2025, doi:10.3390/cells14171301_

Round 1
Reviewer 1 Report
Comments and Suggestions for Authors
This review manuscript by Baudry and Bi summarizes the progress made in understanding calpain-1 and calpain-2 function in the brain over the past 45 years. Structuring the content around distinct eras to highlight major discoveries is a creative approach; however, a few inconsistencies should be addressed to improve clarity and coherence.
To begin with, the introduction references findings from the "last 60–70 years," which does not align with the "45 years" stated in the title and throughout the manuscript. This inconsistency should be corrected. Additionally, the final section lacks a proper ending, as it does not conclude with a period. There is also a formatting inconsistency across sections: in the first section, data are presented first, while in subsequent sections, data are presented at the end. To enhance readability and flow, it may be beneficial to remove the era-based headings altogether, especially given the potential overlap of findings across time periods.
The figures should be improved to better support the text. Specifically, the authors are encouraged to include:
- A more detailed depiction of calpain structure, mechanism of action, and inhibition
- A clear schematic illustrating the overlapping roles of calpain-1 and calpain-2, including their mechanisms of action and cellular outcomes
Overall, more descriptive and well-organized figures and diagrams are necessary. Several mechanistic details that could help clarify the role of these enzymes in neurogenesis, signaling, and inhibition are currently lacking. The manuscript would also benefit from a more cohesive flow of content, and a conclusion, summary, or perspective section is currently missing.
Additional recommendations for improvement:
- The manuscript contains extensive jargon and numerous undefined abbreviations. For instance, on page 4: TBI and LTP; page 6: BDNF and NMDA; and page 7: nNOS, PHLPP1, PDZ. These should be clearly defined upon first use.
- Although the review is intended to focus on calpain-1 and calpain-2, other isoforms such as calpain-3 and calpain-9 are mentioned (e.g., on page 5) without a clear rationale or connection. Their inclusion should be clarified or limited.
- Since calpain-1 and calpain-2 have opposing functions, any reference to general "calpain inhibitors" should be carefully qualified, particularly in Section 3.
- Figure 1 needs revision. In Panel A, calpain-1 should be labeled directly. Consistent shapes should be used to represent proteases (e.g., the "Pac-Man" vs. circle discrepancy should be resolved). Additionally, the schematic does not clearly demonstrate how calpain activation or inhibition leads to changes in signaling or protein interactions.
- More detailed information about enzyme mechanisms and inhibition should be included to aid reader understanding.
- In reference to the MD simulation data on page 10, a structural figure highlighting key regions such as the P1' and P3 pockets would significantly enhance clarity.
- In Figure 2, the authors compare calpain-1 deletion with calpain-2 inhibition, yet the effects of calpain-1 inhibition are not addressed. For consistency and clarity, it may be more effective to present calpain-1 and calpain-2 on the same axis, varying only by activation or inhibition.
- The manuscript contains a large number of self-citations. While not inherently problematic, the authors should ensure they have thoroughly reviewed and cited relevant work from other groups.
- The use of first-person language such as “we did this” should be replaced with more objective phrasing, referring to the specific research group or laboratory.
- A dedicated section titled "Conclusion" or "Perspective" is needed. The authors could expand on the brief paragraph currently at the end to provide a summary, future outlook, and discussion of emerging directions. A brief mention of alternative strategies, such as protein or peptide-based selective calpain inhibitors, would be valuable.
- There are numerous minor issues that should be addressed, including misaligned figures, inconsistent formatting, and incomplete or missing information in author affiliations, grant numbers, and references.
In summary, while the manuscript provides a valuable historical overview of calpain research, the presentation would benefit from structural revisions, clearer figures, better organization, and a more comprehensive and balanced perspective.
Reviewer 2 Report
Comments and Suggestions for Authors
The work by M. Baudry and X. Bi is a review about the roles played by calpain-1 and calpain-2 in the brain. It is very well-written and will be very useful to the scientific community. Please see below a couple of additions to be made:
-The chemical structure of NA-101 inhibitor should be shown in Figure 3 for a better understanding of the content of the manuscript.
-A new figure including a snapshot of the NA-184 inhibitor into the active site of calpain-2 would be very helpful to the scientific community.
Round 2
Reviewer 1 Report
Comments and Suggestions for Authors
Most of the comments were addressed. There is still referring to their lab s previous work as "we" in multiple instances. For example, p.10 "Although the first selective calpain-2 inhibitor we identified, NA-101 (Fig. 4)", or p. 8 "These results suggest that calpain- 342 1 plays a positive role in neurogenesis. In a very recent study, we showed that the tran- 343 scription factor, MEIS2 (Fig. 2B), is a target of calpain-2 but not of calpain-1. ". Keep this informations consistent with rest of the text.
Author Response
Most of the comments were addressed. There is still referring to their lab s previous work as "we" in multiple instances. For example, p.10 "Although the first selective calpain-2 inhibitor we identified, NA-101 (Fig. 4)", or p. 8 "These results suggest that calpain- 342 1 plays a positive role in neurogenesis. In a very recent study, we showed that the tran- 343 scription factor, MEIS2 (Fig. 2B), is a target of calpain-2 but not of calpain-1. ". Keep this informations consistent with rest of the text.
We have modified the text as suggested.